# Integrating Social Care into Healthcare: A Review on Applying the Social Determinants of Health in Clinical Settings

**DOI:** 10.3390/ijerph20196873

**Published:** 2023-10-02

**Authors:** M. Lelinneth B. Novilla, Michael C. Goates, Tyler Leffler, Nathan Kenneth B. Novilla, Chung-Yuan Wu, Alexa Dall, Cole Hansen

**Affiliations:** 1Department of Public Health, Brigham Young University, Provo, UT 84602, USA; tylerleff@gmail.com (T.L.); nknovi77a@gmail.com (N.K.B.N.); kevinwu900624@gmail.com (C.-Y.W.); alexaashlidall@gmail.com (A.D.); colehansen1135@gmail.com (C.H.); 2Harold B. Lee Library, Brigham Young University, Provo, UT 84602, USA; michael_goates@byu.edu

**Keywords:** social determinants of health, SDH, SDOH, clinical setting, clinical integration, clinical application

## Abstract

Despite the substantial health and economic burdens posed by the social determinants of health (SDH), these have yet to be efficiently, sufficiently, and sustainably addressed in clinical settings—medical offices, hospitals, and healthcare systems. Our study contextualized SDH application strategies in U.S. clinical settings by exploring the reasons for integration and identifying target patients/conditions, barriers, and recommendations for clinical translation. The foremost reason for integrating SDH in clinical settings was to identify unmet social needs and link patients to community resources, particularly for vulnerable and complex care populations. This was mainly carried out through SDH screening during patient intake to collect individual-level SDH data within the context of chronic medical, mental health, or behavioral conditions. Challenges and opportunities for integration occurred at the educational, practice, and administrative/institutional levels. Gaps remain in incorporating SDH in patient workflows and EHRs for making clinical decisions and predicting health outcomes. Current strategies are largely directed at moderating individual-level social needs versus addressing community-level root causes of health inequities. Obtaining policy, funding, administrative and staff support for integration, applying a systems approach through interprofessional/intersectoral partnerships, and delivering SDH-centered medical school curricula and training are vital in helping individuals and communities achieve their best possible health.

## 1. Introduction

“Conditions in which people are born, grow, live, work, and age” [1,2] impact health. These living and working conditions are created by multi-level socioecological structures, systems, policies, and processes that influence behavior, health, quality of life, and the risks for disease, disability, and death [1,2,3,4]. The confluence and interactions of socio-economic, political, and environmental factors, including discrimination and racism, create social milieus that impact health. Collectively, they comprise the social determinants of health (SDH) [1,2,3,4]. 

Social conditions and health are interconnected. The day-to-day social circumstances create social patterns of health, disease, disability, and death that are disproportionately experienced by individuals and communities, particularly those who are vulnerable. These social drivers are the root causes [1,2,3,4,5] of inequities in health—the unfair, systemic, and avoidable differences in health status and life expectancies [3], which are most pronounced among the vulnerable, at-risk, and underserved [1,2,3,4,5].

Social inequities cause health inequities. Health is an indicator of social and economic progress. Hence, where social and economic inequities exist, health inequities follow [5,6]. The patterns of inequities seen within and between countries manifest as gradients in health outcomes according to social class [1,2,6]. Social position comes with opportunities, choices, and power over resources. Thus, health worsens as one’s socioeconomic position declines [1,2,3,4,5,6]. The reverse is true. Health improves as one’s social position advances in society [6]. 

Social factors that influence health are multiple, complex, interrelated, and dynamic. The degree of social and health inequities is driven by both structural and intermediary determinants [1,2,3,4,5,6]. The structural drivers of health are the macro-level factors created by “how we manage our affairs in society” [5] such as governance, policies, norms, and societal values [1,2]. Together, they dictate social stratification according to education, occupation, income, gender, and race/ethnicity. On the other hand, the intermediary determinants of health are the daily living and working conditions and early life experiences based on one’s social standing. Interacting with behavior, lifestyle choices, material resources, and psychosocial support, day-to-day circumstances shape risks, exposure to disease, and access to healthcare [1,2]. 

There are several social variables that indicate social and health inequities. Hillemeier et al. (2004) categorized the social determinants of health into a core set of 12 domains according to the socioeconomic characteristics of communities as gleaned from local datasets: (1) economic; (2) employment; (3) educational; (4) political (political participation and factors influencing public policy); (5) environmental (air, water, and noise pollutants, environmental hazards, hazardous waste, heavy metals, pesticides, climate extremes, physical design and safety structures in communities); (6) housing; (7) medical (availability and delivery of healthcare services); (8) governmental (features and functions of local governments); (9) public health; (10) psychosocial (aspects of social capital); (11) behavioral (tobacco use, physical activity, diet/obesity, alcohol, illicit drug use, violence); and (12) transport (health-related aspects of transportation) [7]. In contrast, Healthy People 2030, a United States (US) Department of Health and Human Services (HHS) initiative that maps out the 10-year national plan for addressing key public health priorities and challenges, has classified the social determinants into five domains: (1) economic stability; (2) education access and quality; (3) healthcare access and quality; (4) neighborhood and built environment, and finally, (5) social and community context [8]. The American Medical Association (AMA) follows the same five-domain SDH classification, which are not typically evaluated in routine patient assessments [9]. But rather than focusing on how discrete determinants are categorized, AMA encourages physicians to look at the bigger picture—the interactions of multiple social factors and their impact on health [9].

### 1.1. The Lay of the Land: Social Determinants of Health 

Medical care is not the major influence on health [10,11,12]. Only 10–15% of health outcomes [13], possibly even less [14], are attributed to medical care, or the organized provision of services to diagnose, treat, restore, and maintain health, or ameliorate injuries and disabilities. The collective impact of factors outside hospital settings, like health behavior (36%), living and working conditions (24%), and environment (7%), have a greater bearing on health than healthcare itself [4,9,14]. For example, McGinnis and Foege [15] found that almost half of the major causes of mortality in the US had underlying behavioral elements: smoking, poor diet, physical inactivity, and alcohol consumption—a finding re-affirmed by Mokdad et al. a decade later [16]. These behavioral risk factors are not innate. Instead, they interact with the social environment [17,18] and are thus modifiable and preventable. For example, the epidemic of obesity has spurred debate regarding lack of self-control versus environmental influences. Brownell et al. argued that the most upstream cause of obesity in our society was a “toxic food environment” that promoted calorie-dense, nutrient-poor food coupled with policies that did not account for the impact of social factors on individual choices [17]. Physiologically and psychologically, humans are wired to respond even to subtle environmental cues [17]. The chronic exposure to food deserts and the bombardment of food and marketing advertisements can eventually erode one’s willpower [17]. Thus, Brownell recommended using “collective action”, the intersectoral collaborative efforts to support “personal responsibility”, such as advocating for healthy food and working with policymakers on calorie labeling and taxing of sugar-sweetened beverages [17]. Doing so can create environments where healthy choices are easier for children and adults. 

Similarly, education and food insecurity influence health outcomes. Education is a social determinant of health that influences access to health services as a function of income, employment, food availability, and housing security. In the United States, access to healthcare is largely tied up with employment, and is therefore contingent on one’s level of education. In 2021, private health insurance comprised 66% of coverage, of which 54.3% was employer-based. On the other hand, public insurance covered only 35.7% of the population, coming mainly from Medicaid at 18.9% and Medicare at 18.4% [19]. This gap in coverage left 27.2 million (8.3%) Americans without health insurance [19].

The education advantage cuts across race and ethnicity. Obtaining a four-year college degree differed by ethnic groups. According to the US Census Bureau, in 2021, 61% of Asians, 25 and older, had bachelor’s degrees, while only 42% of Whites, 28% of Blacks, and 21% of Hispanic adults were college-educated [20]. When asked about the reason for not completing a bachelor’s degree, more than half of Hispanic adults (52%) claimed being unable to afford a college education, and both Hispanic and Black adults cited needing to set aside their education to work and support a family [20]. 

Education exposes the health gap. Those with a high school diploma and no further education experience poorer health and shorter lives compared to those with university degrees [6]. The difference in earnings continues to widen according to the level of education—with a USD 20,000 difference or more in earning capacity between US college graduates and those with a high school diploma [20]. Further, having college-educated parents provides income and wealth advantages. Households with college-educated parents have a higher median annual income of USD 135,800 with a median wealth of USD 244,500. Conversely, the median income and median wealth for first-generation college households were only at USD 99,600 and USD 152,000, respectively—a staggering difference of more than USD 36,000 on income and USD 92,500 on wealth [21]. Additionally, first-generation college students come from less financially secure households, and thus, are more likely to incur an education debt to finance their studies [21]. Overall, higher education is a health advantage. As a measure of social position, it opens the door to greater opportunities. It correlates not only with personal success, better employment, and higher earnings, but also with a healthier and longer life.

Food insecurity is a social determinant of health. It affects more than 34 million Americans, including 9 million children [22]. It is a “household-level economic and social condition of limited or uncertain access to adequate food” [23]. In contrast, “hunger” is a physiological condition that may result from food insecurity [23]. According to the US Department of Agriculture (USDA), food insecurity may be “low” or “very low”, with “very-low” defined as the reduced intake and interruption of normal eating patterns due to financial and resource challenges [23]. 

As indices of material hardships, food insecurity and housing instability are traditionally subsumed under key socioeconomic metrics and may not always be screened for. However, these specific needs may signal the initial signs of deeper problems such as unemployment/under-employment and multi-generational poverty. Moreover, the uncertainty of food supply, living in substandard housing, and the possibility of evictions create psychosocial stressors that can negatively impact health, including family functioning and interactions [24,25]. 

Food insecurity and housing instability are determinants of chronic diseases, childhood developmental issues, adverse childhood experiences (ACEs), and multigenerational poverty. Being food-insecure is associated with cardiovascular disease [26], poor control of blood sugar [27], and delays in seeking and following treatment [24,28,29]. To illustrate, almost 45% of deaths from heart disease and diabetes are attributable to diets high in sodium and processed meat, but low in fruits, vegetables, nuts, and seeds [26]. Additionally, Mahmood et al. [28] found that 50-to-74-year-old women experiencing food insecurity were 54% less likely to get a timely mammogram in the past two years—representing missed prevention opportunities that place women at an increased risk due to diagnostic delays. The barriers to timely breast cancer screenings were a function of the social determinants of lower education, lower income, lack of access to care, food insecurity, and the lack of awareness about cancer screening guidelines [28]. Conversely, timely breast cancer screenings were associated with higher education, higher household income, and regular access to healthcare and advice [28]. 

Food insecurity is tied to housing instability. The high costs of housing prevent families from meeting the most fundamental needs. Low-income households are forced to make difficult choices, like the decision to “treat or eat”, in which basic needs, like food and access to care, are traded off for high housing costs [30]. Housing instability consists of challenges with paying rent, overcrowding, frequent change of residences, or spending most of the household income on rent or mortgage [24,31,32]. Even before the COVID-19 pandemic, 20.4 million US households already used more than 30% of their income for rent [33]. In 2020, a large percentage of households that were behind on rent also experienced food insufficiency [33]. This was true for Whites (53%), Blacks (47%), and Hispanics (45%) [33]. 

Depending on the amount households spend on rent or mortgage, they may be moderately cost-burdened (>30% monthly income used on housing) or severely cost-burdened (>50% monthly income used on housing) [34]. With most of the income spent on rent, cost-burdened households have scarcely anything left for food, clothing, utilities, and healthcare [33]. These households were more likely to be Black (54%) and Hispanic (52%) [33]. Low-income families with children were also hardest hit by housing challenges. In 2020, low-income households with children under 18, with moderate cost burdens, spent 57% less on healthcare and 17% less on food while those with severe cost burdens spent 93% less on healthcare and 37% less on food [33]. To illustrate further, for households with incomes between USD 15,000 and USD 30,000 in 2019 which were moderately cost-burdened, USD 1150 was left each month after rent [33]. Severely cost-burdened families in the same income bracket had far less—only USD 600 remained after rent was deducted [33]. On the other hand, households who made less than USD 15,000 in 2019 and were moderately cost-burdened had USD 550 left each month, while severely cost-burdened households barely had USD 225 left—an amount clearly insufficient to cover all other expenses, let alone food and healthcare [33]. As families are pressed to find housing in high-poverty and unsafe neighborhoods, women in these families are forced to socially isolate to avoid street danger and violence, which can result in the unintended increase in asthma, depression, and stress [35].

### 1.2. “Moving Upstream” in Healthcare through the Triple Aim 

What is happening upstream has downstream effects. The classic public health parable attributed to Irving Zola [36] illustrates how societal-level root causes lead to individual-level consequences. Zola’s story begins with a rescuer who saved a man from drowning downstream. Not long after this heroic deed, he heard other cries for help. Several people were caught in the fast-moving current and were drowning. He rescued and revived as many as he could. But the situation kept recurring. He became preoccupied with “jumping in, pulling [people] to shore, [and] applying artificial respiration” that there was hardly any time to find out what was happening upstream and why people were falling down the river in the first place [36].

Zola used the terms “upstream” and “downstream” to define the points at which we can intervene to save lives. What is happening upstream are the “causes of the causes”, [1,2]—the structural determinants of governance, laws, macroeconomic policies, and cultural values that create social stratifications and conditions midstream [1,2,37,38]. When unmet and/or ineffectively addressed, individual social needs and risks lead to the downstream consequences of chronic diseases, mental health issues, and behavioral conditions, leading to disability and/or premature death [37,38]. The net effect is poor population health. 

This river analogy classifies not only the nature but also the most effective direction of prevention efforts [38]. Saving those who have fallen downstream is both a moral and an ethical imperative. However, the disproportionate focus on “rescue care” distracts from prioritizing and pursuing “preventive care”, i.e., moving “upstream” to save even more lives by tackling the social determinants of health [1,2]. Most importantly, Zola’s analogy is a call for public health and healthcare to pause and re-examine priorities and resources to help patients and communities “realize [their] birthright of health and longevity” [39]. 

Berwick, Nolan, and Whittington, of the Institute for Healthcare Improvement (IHI), introduced a roadmap to address the inequities in health, and thereby, improve healthcare in the United States—the Triple Aim [40]. This conceptual framework emphasizes the simultaneous achievement of three interrelated and reinforcing goals: “improving the patient experience of care (including quality and satisfaction)”, “improving the health of populations”, and “reducing the per capita cost of healthcare”—in other words, “better care, better health, at lower costs” [40,41]. Additionally, the Triple Aim bridges the gap between public health and healthcare by linking clinical care with community-based prevention strategies [42]. This framework was subsequently adopted in 2010 as part of the national healthcare strategy in implementing the Patient Protection and Affordable Care Act (ACA) [42]. A fourth aim, proposed in 2014 by Bodenheimer and Sinsky [43], addresses burnout (exhaustion, cynicism, and professional dissatisfaction) among healthcare workers—a situation exacerbated by the COVID-19 pandemic. This was expanded in 2022 into a Quintuple Aim as suggested by Nundy, Cooper, and Mate [41] with the addition of “workforce well-being and safety” and “advancing health equity”, with health equity as the ultimate aim. Proponents of the Quintuple Aim argued that the Triple Aim could not be achieved without these additional aims, let alone, health equity [41].

The social determinants of health inequities are “wicked” problems. They are multiple, closely linked, interdependent, and economically demanding, but their full impact and mechanisms are yet to be fully understood—making them challenging, if not impossible to be addressed solely by the healthcare sector. Instead, these require a whole-of-government approach [44,45]. Given the complexity of interactions within and between social factors and health, the Marmot Review [6] calls for action on six policy areas, which have implications on integrating social care into healthcare: (1) “Give every child the best start in life”; (2) “Enable all children, young people and adults to [maximize] their capabilities and have control over their lives”; (3) “Create fair employment and good work for all”; (4) “Ensure healthy standard of living for all”; (5) “Create and develop healthy and sustainable places and communities”; and (6) “Strengthen the role and impact of ill health prevention”. 

The purpose of our study was to contextualize the present state of SDH integration in US clinical settings. Through an objective and critical analysis of the existing literature, we sought to answer the following research questions: (1) What clinical translation strategies for incorporating SDH into routine clinical practice were identified in the literature? (2) In which patient population demographics and health conditions were SDH clinical assessments most frequently applied? (3) What were the primary reasons for integrating SDH in clinical settings? (4) Which SDH screening tools were used in clinical settings? and (5) What were the challenges, lessons, and recommendations in integrating SDH in clinical settings [46]? In addition, we noted the types of US healthcare facilities that integrated SDH in clinical settings, including their geographical locations, when indicated. We also documented the current efforts of the healthcare sector in integrating social care into healthcare to guide present and future endeavors in providing equitable care.

## 2. Materials and Methods 

We conducted a narrative literature review to explore the ways in which the social determinants of health were being addressed in clinical practice. Articles were included in this review if they met the following criteria: (1) written in English; (2) studies focused on geographical locations within the United States; (3) discussed the application of SDH in clinical settings in the United States; and (4) published from 1 January 2008 to 31 March 2023. We used 2008 as the publication start date of included article to coincide with the final report of the World Health Organization (WHO) Commission on the Social Determinants of Health [1], the date range of which would be inclusive of the 2010 passage into law of the Patient Protection and Affordable Care Act (ACA) [42]. To identify relevant articles, we conducted an initial literature search in PubMed on 16 February 2023, and an updated search on 31 March 2023 for English-only articles published from 1 January 2008 to 16 February 2023, using the following search terms:

(social determinants of health OR SDH OR SDoH OR social factors) 

AND (primary care OR healthcare OR hospital OR clinic) 

AND (screening OR screen OR policy OR triage OR patient outcome)

AND (America OR US OR USA OR United States OR American)

These searches yielded peer-reviewed empirical studies and theoretical papers, such as literature and/or policy reviews, commentaries, and clinical opinions on SDH in US clinical settings. The retrieved articles were initially screened by two researchers to ensure that they addressed how clinicians were incorporating social care in US clinical settings. Seven researchers subsequently conducted full-text reviews of all included articles. The following data were extracted from each of the reviewed articles: Study methodology;Clinical setting where SDH was being addressed;Patient health outcomes of interest;Ways in which SDH was integrated into clinical practice;Specific SDH screening tools used in clinical practice.

The principal investigator summarized all the articles and created tables in Word for reporting the findings of the narrative review according to the stated aims of the study. Three other researchers assisted the principal investigator in tabulating the information gleaned from the included articles according to the following categories: Study design and methodology;Publication year;Target patient population and medical conditions for which SDH assessment was conducted;Healthcare providers and facilities involved in SDH integration;Strategies/methods for SDH integration;SDH screening tools used in clinical settings;Primary reasons for SDH integration;Barriers, facilitators, and recommendations from the literature on how to successfully apply SDH in clinical settings.

The information on the tables were then re-reviewed and edited by the principal investigator with the help of one of the researchers.

## 3. Results 

The articles selected for inclusion in this review were identified using PubMed, which resulted in a total of 194 items published from 1 January 2008 to 31 March 2023. Following an initial review, 44 articles met the inclusion criteria. A full text review of each of the 44 articles was conducted followed by team discussions based on the inclusion criteria and research questions. Of the total 44 articles included in this literature review, 19 were quantitative studies, 9 were qualitative studies, 5 were mixed methods studies, 2 were systematic reviews, and 9 were theoretical papers such as literature and/or policy reviews, commentaries, and clinical opinions. More than half of the empirical studies (66%) included in this paper were published from 2020 to 2023, 10 to 13 years from the passage of the comprehensive healthcare reform (ACA) into law [42], while theoretical papers were more common on or before 2020, or within the first 10 years of ACA. See Table 1. The subsequent tables present data to address the study objectives as identified at the end of the introduction. Specifically, Table 2 provides information on patient population demographics and health conditions underpinning SDH clinical assessments, Table 3 lists various SDH screening tools used in clinical settings, Table 4 identifies various reasons for SDH integration, and finally, Appendix A Appendix A lists the study design of the articles included in this review, method of integration, type of healthcare facility and its geographic location.

### 3.1. Integrating SDH in Clinical Settings as to Target Patient Population, Conditions, and Type of Healthcare Provider 

SDH assessments in clinical settings focused primarily on vulnerable, disadvantaged, at-risk, and/or socially isolated patients. These patients came from communities experiencing persistent socio-economic challenges, material deprivation, food insecurity, housing instability, and/or toxic stress. Incorporating social care into healthcare was frequently carried out in primary care settings within the context of chronic diseases, such as cardiovascular disease, cancer, diabetes, and hypertension; mental health issues, such as anxiety, depression, and ACEs; and behavioral issues such as smoking or substance use disorders, and cancer screening behavior for breast, colorectal, cervical, or prostate cancer.

The next most common SDH screenings occurred in pediatric settings. Through voluntary self-reported surveys, parents and caregivers provided information regarding their family’s social needs and how necessities, such as food, housing, and transportation, affected access to care, duration of hospital stay, and the severity of their child’s medical condition. This offered the advantage of obtaining information on the family, as a whole, and understanding the situation they were going through as a collective unit. See Table 2.

### 3.2. Integrating SDH in Clinical Settings by Study Design, Method of Integration, and Type of Healthcare Facility

The most common method documented in the literature for integrating SDH in clinical settings was through the use of SDH screening tools. These were typically administered as self-reported surveys or questionnaires during patient check-in and were often used in conjunction with a community referral program. These tools varied in length and in the SDH domains that they screened for. Of the 44 articles included in this review, 29 different SDH screening tools, including digital referral platforms, were cited in the literature. These 29 tools were available for use, actually used, or recommended for use in clinical settings. The 26-item 13-domain Accountable Health Communities Health Related Social Needs (AHC HRSN) screening from the Centers for Medicaid and Medicare Services (CMS), was the most commonly cited screening tool, having been mentioned in six out of 44 articles (13%) [56,69,76,81,86,87]. CMS intends to use this tool as the standard screening tool to assess social needs in at least seven million Medicare and Medicaid recipients being served by all participating healthcare facilities in 32 AHCs across the nation. The original tool consisted of 10 questions in five core domains [88]. CMS has since added 16 questions on eight supplemental SDH domains to offer AHCs flexibility in using any of the eight supplemental domains in their standard screening process [88].

The next three most common screening tools were the Hunger Vital Sign (HVS), Protocol for Responding to and Assessing Patient Assets, Risks, and Experiences (PRAPARE), and the Safe Environment for Every Kid Parent Questionnaire-R (SEEK PQ-R). The two-item, single domain HVS, is a validated tool that has been used in medical and community settings nationwide for assessing food insecurity in families with young children, adolescents, and older adults [26,68,71,75]. PRAPARE is a 21-item, 21-domain, validated, and nationally standardized tool that evaluates 21 social drivers of health in five core categories, and is designed for adult respondents. Although traditionally utilized by health centers, it can also be used by hospitals, healthcare systems, and community-based services and organizations [56,57,76,81,89]. The 16-item, multiple-domain SEEK PQ-R is a validated primary care screen for child abuse and neglect that is voluntarily completed by parents or caregivers before their child’s appointment [51,67,76,87,90].

Two digital interactive tools were mentioned in the articles: the American Academy of Family Physicians’ (AAFP) Neighborhood Navigator, which is a point-of-care connection with community resources within the patients’ own neighborhoods [48,86], and NowPow, an electronic referral platform that allows providers to link patients and track referrals with a wide range of community resources within their respective zip codes [57,69,91]. NowPow, which can be integrated into EHRs and health information exchange (HIE) systems, also provides visualizable analytics on the pattern and frequency of referrals for specific social needs, such as food insecurity, housing instability, and/or transportation issues [91]. See Table 3.

Various qualitative methods of data collection were mentioned in the literature. Qualitative studies in this review utilized surveys, focus groups, and semi-structured interviews of clinical leaders, providers, frontline staff, and patients to assess their perspectives on the development and/or implementation of SDH screenings. These methods were used either solely or in conjunction with quantitative analyses to obtain richer data. For example, Okafor, Chiu, and Feinn’s (2020) study on food insecurity in a pediatric practice utilized both the two-item HVS survey and a focus group interview of pediatricians to identify the challenges with universal SDH screenings in pediatric settings [75]. Patient interviews provided valuable information regarding their hesitancy and discomfort with SDH screenings. Patients expressed concern over the stigma and bias of being food-, job-, and/or housing-insecure [75] and feared being discriminated against obtaining community resources [51]. On the other hand, physicians shared their insights on the challenges of implementing SDH screenings, such as time constraints, and the difficulty of incorporating SDH screenings in patient workflows and EHRs [51,70,78]. 

Other study designs were noted in this literature review. Mixed methods studies, systematic reviews, and theoretical papers captured more nuanced perspectives on the feasibility of integrating SDH assessments in clinics and hospitals. Overall, the value of SDH screenings in clinical settings and their impact on patient health was generally met with positive feedback from providers and patients. 

Primary care facilities were most commonly involved in conducting SDH assessments in clinical settings. These included primary care and pediatric ambulatory clinics, federally qualified health centers (FQHCs), and safety net hospitals, particularly those serving urban communities. See Appendix A Appendix A.

### 3.3. Major Reasons for Integrating SDH in Clinical Settings 

Several reasons were cited in the literature for integrating SDH in clinical settings. Foremost among these reasons was to implement SDH screening processes or programs to identify unmet social needs, evaluate risks, and link patients to community resources (26 articles). This was followed by conducting formative and process evaluations of existing clinical SDH screening processes or programs, which were supplemented with clinical performance measures in some studies [67] and/or provider and patient feedback [67,69] for a more efficient integration into patient workflows (15 articles). Only seven articles discussed evaluating hospital-level community outreach efforts, particularly for non-profit or safety net hospitals [47,50,52,65,72,77,79]. Additional reasons for integration focused on reducing overall health inequities in hospital processes [74,83]; improving current methods to more effectively incorporate SDH in clinical settings [18]; and enhancing the training of future healthcare professionals in identifying and addressing SDH needs and risks [71]. Improving cost-efficiency within the healthcare system was another primary reason for SDH integration in some studies, but which played a secondary role to improving population health—a duality of purpose described by Brewster et al. (2020) as “improv[ing] health and thereby reduc[ing] unnecessary healthcare use and spending” [52]. See Table 4.

Applying SDH in clinical settings was either a short-term or a long-term endeavor. The former referred to temporary applications, typically performed for research purposes, while the latter referred to systems that have already adopted relevant SDH practices. The short-term SDH integration in clinical settings was typically motivated by specific research questions to determine the impact of social factors on various clinical outcomes. For example, some articles included in this review explored the role of the social determinants of health on the following outcomes: exacerbating depression and reduced prostate cancer screening behavior [50]; lowering the risk for unintentional injuries among children aged 1 to 5 years [67]; and initiating screening and medications among rural veterans who were at risk for osteoporosis [59]. Although some studies started as short-term, proof of concept investigations, in reality, they were exploratory pursuits into the long-term feasibility of universal SDH screenings in routine clinical practice [54,75]. For example, Selvaraj et al. (2019) investigated the feasibility of a universal screening of the social factors associated with ACEs and its acceptability in a medical home in which 86% of the 446 families surveyed expressed their desire for the clinics to continue the screening [53].

## 4. Discussion

### 4.1. Factors Promoting the Clinical Translation of SDH and Their Implications 

External and internal factors facilitated SDH screening in clinical settings. External factors included grant requirements [70], encouragement from professional organizations and stakeholders [56,63,70], engagement with community partners [46,50,54,79], state and federal policies and laws [18,46,50,65,77,85], and third-party payer motivation [18,48,52,56,65,77,81]. Of these external factors, third-party payers were identified as one of the most important motivators to SDH implementation. Multiple studies identified how third-party payers, such as CMS, exerted a positive influence on SDH implementation in hospitals and other medical institutions. In particular, CMS developed financial incentives to reduce ED visits and hospital readmissions [65]; created waivers to cover SDH needs [77]; reimbursed telehealth visits at regular clinic visit rates [81]; and covered patient transportation expense to medical appointments [81]. These third-party payer efforts exercised significant influence on healthcare organizational behavior toward SDH. However, additional research is needed to identify how third-party payer initiatives impact overall community health and well-being.

Other important external motivators identified by multiple studies in this review were federal and state policies and laws. Of particular note was the impact of the 2010 Patient Protection and Affordable Care Act (ACA), specifically the requirements for tax-exempt hospitals to conduct community health needs assessments (CHNA) every three years [46,65,77]. In addition to mandating CHNAs, the ACA also required tax-exempt hospitals to report on an implementation strategy for addressing the social needs in the communities that they serve [46,65]. To fulfill this requirement, hospitals frequently engaged in charity care, housing initiatives, and other community-building activities [18,46]. Additionally, state-level policies exerted a profound influence on how healthcare facilities address SDH in their communities. For example, hospitals located in states with more robust laws regarding prescription pain medications were more likely to implement strategies to address opioid misuse in their communities [50]—highlighting the influence that policymakers wield over healthcare organizations’ efforts in addressing social factors in their communities. As such, public health practitioners and researchers need to carefully consider the ways in which they can engage policymakers in developing and implementing policies that will improve health outcomes across communities.

Engagement with community partners was an important predictor of a healthcare organization’s involvement in SDH [46,50,54,79]. One possible explanation for hospitals’ increased involvement with community partners is the recognition of their expertise in addressing local SDH issues [50], which would allow hospitals to effectively reach specific communities [79]. This underscores the importance of fostering collaborative relationships with community organizations to effectively address social needs, especially among diverse populations. Moreover, public health practitioners are uniquely qualified to foster coalitions between healthcare organizations and community partners. Public health organizations can assist in identifying community partners that can serve as potential collaborators with healthcare organizations in developing initiatives that will best meet the health needs of the community. 

Several internal factors facilitated SDH implementation. These included organizational factors such as affiliation with a larger health system [52] and designating a SDH screening advocate or team within the organization [70]. Other internal factors included employee training [54,72,83], workplace flexibility [70], adopting technological advances [46,48,69,85], and implementing appropriate screening tools [46,69,76]. One study identified the value of participating in an externally certified project to ease the implementation of SDH screening and referral [66]. Similarly, developing a smaller-scale pilot project to address social needs in a clinical setting can serve as the first step in developing a sustainable SDH initiative [57]. The wide variety of internal factors identified in the literature emphasized the importance of developing an organizational culture that is flexible and willing to apply innovative ways in addressing social needs. Developing a training program for healthcare staff that explicitly identifies the impact of SDH on community health outcomes can enhance employee participation in SDH-centered healthcare initiatives. Additionally, healthcare organizations with an appointed person or team responsible for SDH implementation are more likely to succeed in engaging the community. Such individuals or teams can lead out in identifying, creating, and applying appropriate SDH screening tools, training staff in assessing patient SDH, and in engaging in community outreach efforts. 

### 4.2. Challenges in Integrating SDH in Clinical Settings 

The integration of SDH into existing workflows was fraught with challenges at various points: educational, practice, and administrative/institutional levels. At the educational level, studies identified the lack of formal SDH training for healthcare providers, that limited their ability to evaluate patient social needs [18,46]. Hudon et al. (2023) pointed out the variations in the delivery of medical curricula and the disproportionate focus on the biology and treatment of diseases versus accounting for the impact of social drivers on clinical outcomes [46]. As such, future physicians lacked the SDH knowledge and competencies necessary to recognize social needs and to adequately refer patients to community resources and services [18,46,73,74,82,86]. 

At the practice level, internal operational challenges created barriers to SDH implementation. Clinicians’ busy schedules inhibited their ability to make SDH a priority in their practices [18,46,54,66,73,82,85]. The limited time but increased demand on clinician’s time subsequently resulted in the lack of opportunity to conduct screenings or to discuss social needs and resources with patients [54,67,68,75]. Additional operational challenges included workflow disruptions from screenings [54,69,76]; workload demand of screenings on an already burdened staff [57,69,73]; lack of staff; and the need for social workers, volunteers, or patient navigators/lay health educators to assist with SDH screenings and referrals to avoid compromising workflow efficiency [51,54,57,63,71,79,82,84,85]. Further, there was a general lack of provider referral skills [75] and familiarity with available community resources and organizations to refer patients to [46,54,81,85].

EHRs created additional challenges with SDH integration. Several studies found that SDH needs were either inconsistently documented or could not be documented on electronic medical records (EMRs) and/or electronic health records (EHRs). Such technical limitations precluded integration of SDH screening information into EHRs nor for referrals to be sent and tracked. Thus, making it difficult for staff to close the loop on referrals [52,54,66,76,81,85]. 

At the organizational/institutional level, the volume-based, fee-for-service model remained a barrier to reaching out to vulnerable populations. Such model incentivizes overutilization of services, which drives up the cost, at the expense of quality, thus, making care less accessible for those that need it most. Until the introduction of the 10 SDH primary Z codes (Z55.0–Z65.0) by CMS, clinicians were not reimbursed for time spent in addressing patient social needs [18,81]. Data variability, quality, confidentiality, and access issues prevented providers from obtaining information on their most vulnerable patients [46,54,81,85]. Additionally, certain medical specialties preferred to work in isolation and lacked the skills to collaborate with community partners [18,46]. This discouraged interprofessional partnerships that could have allowed for multiple expertise to address the diverse facets of patients’ social and medical needs. Lastly, several clinicians and health systems chose not to participate in SDH interventions because they were uncertain if such interventions will actually improve patient outcomes [18,81].

For healthcare providers who worked with community partners, multiple operational issues were identified. For example, maintaining patient confidentiality was challenging when working with community organizations [46]. Healthcare providers and community partners have different modes of operation that lacked a common vocabulary on medical and social needs [46]. Likewise, healthcare providers needed training—not only on SDH, but also on effective strategies for community collaborations. 

### 4.3. Lessons Learned and Recommendations on Integrating Social Care in Healthcare

The United States spends trillions of dollars on healthcare. In 2021, the total healthcare spending in the US rose to USD 4.3 trillion or 18.3% of the Gross Domestic Product (GDP)—a 2.7% increase that translated to USD 12,914 spending per person [92]. Despite outspending every country in the Organization for Economic Cooperation and Development (OECD) on healthcare, America’s life expectancy of 76.4 years (2021) was comparable to countries like Croatia (76.7) or Colombia (76.9) that spent far lower dollars on healthcare in relation to their GDP [93]. 

The huge investments in healthcare and the soaring cost per capita of care have yet to translate into longer and healthier lives. In 2021, the life expectancy at birth for Americans dropped to its lowest since 1996. From 78.8 years in 2019, before COVID-19, it decreased to 76.4 years in 2021 [94]. Disparities by gender and ethnicity continued to widen. Men (73.5 years) lived almost six years shorter than women (79.3 years), a gap attributed primarily to COVID-19 deaths. In terms of disparities by race and ethnicity, the expected length of life for Native Americans and Alaskans shortened by almost 2 years (1.9 years), the biggest reduction compared to other ethnic groups in 2021 [94].

The major causes of death in the US have persistently been lifestyle-based. Although COVID has taken over the third highest spot, heart disease and cancer have remained as the top two leading causes of death. Chronic liver disease/cirrhosis and suicide have contributed 3% and 2%, respectively, to the shortening of life expectancy. Although 50% of the decline in life expectancy was attributed to deaths from COVID-19, 16% or 106,699 deaths actually came from accidental deaths/injuries, of which almost half were due to drug overdoses, particularly of opioids like fentanyl and methamphetamine [94]. These deaths were mostly among those aged 35 to 44—possibly a reflection of the psychosocial stressors heightened by the pandemic. 

Life expectancy reflects the prevailing mortality patterns across age groups as a function of their environment. On the flip side, it also mirrors overall health. Better living standards, healthier lifestyles, higher education, and greater access to quality health services have been shown to significantly extend life expectancy. For instance, five lifestyle factors have been shown to add 12 to 14 years to one’s lifespan at age 50, for men and women, respectively: never smoking, having a normal body mass index (18.5 to 24.9 kg/m^2^), moderate-to-vigorous physical activity for ≥30 min/day, moderate alcohol intake, and a high-quality diet [95]. The scientific evidence for the impact of behavior and lifestyle on health has been brought to our attention decades ago by McGinnis, Foege, and Mokdad et al.—warranting a serious consideration of preventive care as a national health priority. 

Patterns of health inequities are evident in the conditions where we are born, live, work, and age. Social and economic conditions that are inherently connected to poor health outcomes are deeply rooted in systemic inequities and are worst where and among whom social disadvantage continue to exist in various forms. Our current model of care has long focused on the downstream consequences of ill health, with little attention and resources left to spare in addressing the upstream conditions that give rise to the “causes of the causes” of health inequities. Studies have repeatedly called for the long-term [46,79,81,85] “integration of social needs into mainstream healthcare” [81]. As Carter and Mazzoni (2021) stated, healthcare’s focus is misplaced unless “a patient’s risk factors [are] contextualized by examining the root causes of elevated risk” [83]—echoing Berwick’s call for pursuing better health at the population level and an enhanced quality of care at the individual level, at a lower cost [40]. 

Healthcare is responding to calls to integrate social care into clinical care by identifying the non-medical factors that significantly impact individual and population health.

Efforts and funding for integrating SDH in clinical settings are increasing. Studies affirming the link between social factors and health outcomes, together with SDH-centric policies at the national level, help advance this objective [77]. Below are the key lessons gleaned from the literature on integrating SDH in clinical settings and their implications to practice, policy, and research:Clinical translation strategies for incorporating SDH into routine clinical practice
State of Integration based on the Literature:
○The main strategy for embedding SDH in clinical settings was through patient screenings [46,49,51,53,54,55,56,57,58,63,65,66,67,68,69,70,75,76,87]. These were typically one-time screenings conducted during patient check-in using self-reported surveys and carried out before a scheduled appointment via phone or online portals, or at the point-of care. ○SDH integration in clinical settings offered the opportunity to promote intergenerational health. SDH screenings among pregnant women can improve two-generations-worth of health outcomes while simultaneously strengthening the healthcare system [73,83]. ○SDH screenings helped in detecting specific social needs, such as food insecurity and/or housing instability [28,64,79], which negatively affected cancer screening behavior, chronic disease outcomes, and smoking-related diseases [26,28,63,64,68,75].
Current Gaps:
○Screening for social factors is not part of the current standard of clinical practice. SDH integration in US clinical settings is still in its formative stage despite the 2010 passage of ACA into law [42]. The majority of empirical studies on SDH integration in clinical settings were published in the last three years, from 2020 to 2023, suggesting increasing efforts to address the social determinants of healthcare. ○The literature mentioned only one primary method for integrating SDH clinically, i.e., through patient-level screenings, followed by referrals to community resources and services. However, it is not yet known whether this is the best method for assessing SDH clinically.○Little can be gleaned from the literature on how tackling unmet social needs can be used to stratify patient risks, modify care, predict clinical outcomes, resolve provider reimbursements for time dedicated on SDH-specific work in clinical settings, or whether existing SDH assessments will be funded, integrated, and supported long-term in routine clinical practice. The lack of data on the impact of SDH integration on improved patient outcomes deterred hospitals from investing in upstream SDH activities, which they would have gone through with had concrete evidence been available [77]. ○Physicians and clinical staff may not be reimbursed for their time and efforts in carrying out SDH-related work [81].
Recommendations:
○Modifying the payment model to incentivize quality care and to capture the full range of reimbursable SDH-related work [18,52,81]. The traditional fee-for-service payment model is a barrier to SDH integration in clinical settings by raising costs through increased utilization [26,65,73]. Moreover, it fails to capture the work invested by healthcare providers in quality care, resulting in reimbursement issues [18,52,65,81]. Although the introduction of the ICD-10 Z codes was a step in the right direction [81], until they are tied to insurance payments with clear instructions regarding their use, the adoption and utilization of these codes will be slow.○Providing scientific evidence for integrating SDH within the full spectrum of care. Further research is necessary to determine how patient SDH data can guide the following: (1) establishing best practices in screening; (2) identifying the SDH domains most strongly connected to health; (3) modifying treatment plans; (4) determining the extent in which SDH patient-level data predict patient clinical outcomes. When combined with data from CHNAs and surveys, patient SDH data can guide policy, practice, and research [47,50,65].○Exploring the socio-economic and political context of cancer screening behavior and time-to-diagnosis and treatment through SDH screenings. Advocating for policies mandating food insecurity screens may help reduce cancer morbidity and mortality [28,79].○Providing primary care practices with healthcare analytics dashboards to inform providers of the SDH characteristics of the communities they serve and to help providers make real-time data-driven decisions [46,59].○Testing other SDH integration methods and integrating data into practice [46] to encourage efforts in addressing the structural causes of health inequities. SDH integration efforts noted in the literature were highly concentrated on alleviating patient-level social needs and risks, but lacked the corresponding efforts at the community level to address the system-level factors that were the root causes of health inequities [83]. Limiting interventions at the patient level misses the mark unless efforts and interventions are carried all the way through the socioeconomic conditions that created the health problems in the first place. 
2.Patient population demographics and health conditions underpinning the clinical assessment or study of SDH, types of health provider, healthcare facilities, and their geographic locations within the United States
State of Integration Based on the Literature:
○Priority populations for SDH screenings were individuals and families from low-income households, who were underinsured/uninsured, medically underserved, or were immigrants with low literacy and English fluency, and were from multicultural communities. These included individuals who were in public housing, homeless, migratory and seasonal agricultural workers, elderly, individuals with disability, those experiencing mental health issues or substance use disorder [50], and Medicaid and Medicare recipients.○SDH integration through screenings were conducted in relation to chronic diseases [18,26,49,54,64,67,68,74,75,80,81,84,85,86], mental health issues [49,53,54,75,83,85], and behavioral issues [28,55,56,59,75,79] compared to acute conditions.○Primary care specialties (pediatrics, family medicine, and internal medicine) and facilities (clinic, health centers, academic medical centers, and safety net hospitals) were most commonly involved in integrating SDH in clinical settings, among urban or inner-city residents and communities. The literature was replete with articles citing the role of social workers, CHWs, patient health navigators, or lay health educators, even public health students, in assisting with patient screenings and referrals [51,54,57,63,71,79,82,84,85].
Current Gaps:
○Integrating SDH in clinical settings came with several internal operational issues in embedding screenings in patient workflows: limited time for screenings and visits, as well as competing priorities, including the time-consuming work of documentation.○Primary care specialties are still tacitly considered as the natural fit for addressing patients’ SDH issues, rather than having SDH integration as a part of routine care and as a shared accountability in all specialties.
Recommendations:
○Identifying lessons learned from implementing screenings by engaging physicians and staff in providing their feedback on improving the process [51,54,66,73,74,75,76].○Testing different multi-level strategies for a more efficient incorporation of SDH in clinical settings and to increase the capacity of physician practices for innovation [46,69]—not only in primary care settings, but as a standard in all specialties. Braveman and Gottlieb (2014) recommended having on-site social and legal resources for patients [45].○Expanding policy support and SDH assessments in clinical settings to include patients in economically marginalized rural areas and from middle-class households [18].○Sharing the accountability for care and health equity needs across medical specialties and sectors. One article explicitly emphasized a systems approach in tackling health inequities through a “health in all policies” or a multi-sectoral whole-of-government approach [18].○Considering augmenting clinic, hospital, and ED staff with social workers, case managers, CHWs, or patient navigators to assist in referring patients to resources and social services in the community [18].○Encouraging physicians to seek interprofessional partnerships that will allow several expertise to address multiple facets of an individual’s health [46]○Training physicians to (1) genuinely listen to obtain the trust of patients when screening for social needs, particularly among patients from diverse racial and ethnic backgrounds [49]; (2) develop community referral and interprofessional collaboration skills [46]; and to (3) engage in clinic–community partnerships, and advocacy work [46]. Interprofessional and intersectoral collaborations offer opportunities to address, through collective action, the adverse local social determinants prevalent in communities where patients live, work, and play. ○Providing patients with a handout, tailored to each practice, which lists local and national organizations to help connect families with the resources that fit their needs [67].
3.SDH screening tools and domains assessed in clinical settings
State of Integration Based on the Literature:
○There was heterogeneity in SDH screenings tools. At least 29 different tools were cited, actually used, or recommended for use in clinical screenings of various social needs. These included two interactive digital platforms, AAFP Neighborhood Navigator and NowPow. Some healthcare facilities created their own screening tools by either modifying existing tools or by developing a new one. In the future, more specialized tools will likely be available for use in clinics and hospitals, especially as healthcare facilities generate their own screening tools. ○There was no standardized SDH screening tool for use in clinical settings. Available screening tools varied in the number and type of questions asked; the number and type of SDH domains assessed; and in the timing of assessment. There was no uniform set of SDH domains assessed across healthcare systems and facilities [53,56,74,76,82]. Instead, various clinics and healthcare facilities used different SDH screening tools according to their institutional or research priorities; modified existing tools [57,67], applied multiple tools [51,55,56,67,68,86], or created their own screening tools [54,63,76]. This could be both a strength and a limitation. As a strength, the multitude of available screening tools allows for options that best align with detecting individual-level social needs and/or the SDH priorities of the healthcare facility. ○Screenings were either integrated or unintegrated in EMRs and EHRs, resulting in inconsistent documentation. The tracking and closing of referrals electronically were possible depending on the digital screening platform used [57,60,63,64,66,68,69,70,86].○HVS [26,58,68,75] and SEEK [51,67,76,87] were among the commonly utilized tools mentioned in the literature, reflecting the pervasiveness of food insecurity and of abuse and neglect among pediatric patients.
Current Gaps:
○The multiplicity of screening tools used in clinical settings presents challenges in establishing best practices in terms of patient screening, data adequacy, consistency, and comparability.○Screening for screening’s sake can be a slippery slope that can distract and detract from the aim of health equity.○Food insecurity is not routinely screened for in clinical settings. Notwithstanding, it is a prevalent issue among various patient populations, as are transportation issues and housing instability [28,80]. These can limit access to timely care, delay screening and time-to-diagnosis, and reduce compliance to treatments. 
Recommendations:
○Standardizing SDH assessments to allow for comparability across healthcare facilities. The lack of a standardized approach to screening presents challenges in determining best practices and in establishing consistency in data collection for comparability across various types of facilities. This will allow for continuity of care, funding, and resource allocation.○Determining from a pragmatic context, if a flexible SDH tool, with universal core metrics but optional ancillary domains, may be of greater practical value than a uniform but rigid tool in detecting diverse social needs and risks. This can help guide decisions in standardizing screening tools.○Screening, in isolation of the broader examination of the systems, structures, and policies that perpetuated social and health disadvantage, is counter-productive to achieving health equity. Patient-level SDH data can inform interventions and advocacy strategies for addressing social factors that impact health at the individual and community levels.○Integrating and advocating for food insecurity screenings in pediatric practices [75] and cancer care [29] and promoting screening behavior among those who are food-insecure [28]. Information on local food sources close to physician’s offices [75] can be provided as well as establishing “food pharmacies” for patients where they can be referred to by their doctors and counseled by dietitians [26].○Engaging patients in deciding which community resources and services best fit her/his needs. Substituting physician-patient discussions with screenings due to time constraints and competing priorities risks patient satisfaction and lessens the perceived value of SDH screenings. Training physicians and staff to genuinely listen to obtain the trust of patients in screening for social needs, especially among culturally and racially diverse patients [49].○Linking SDH screenings, referrals, and tracking with EMRs and EHRs to make it easier for the clinical staff to consistently and efficiently document patients’ social needs. Doing so was recognized by the healthcare leadership, providers, and staff as an opportunity to expand research on how SDH integration could affect clinical outcomes, such as disease onset, severity, and length of hospital stay [55,61,80].
4.Reasons for SDH integration
State of Integration Based on the Literature:
○The major reason for integrating SDH clinically was to identify risks and to refer patients to community resources and services. This was noted in 25 of the 44 articles included in this review.
Current Gaps:
○The literature had little information on the unintended consequences of screening on patients, physicians, and staff. Aside from independent facility experiences, little is known at the sectoral level on whether SDH screenings made a positive impact on individual clinical outcomes nor on the long-term utilization of services and the cost of care. ○The literature did not mention what happened beyond SDH screening. It was not always clear from the included studies whether further follow-up was carried out beyond the initial SDH assessment to assess if social needs were being adequately met. 
Recommendations:
○Further research is needed to determine whether addressing the social determinants of health in clinical settings improves patient health outcomes. 


### 4.4. Limitations and Future Research Directions 

This study is based on a review of the current literature on SDH in clinical settings. Although an objective and critical appraisal of current studies, it is not an exhaustive nor a systematic review of the literature. Thus, it is possible that we have missed other articles relevant to our study. For instance, there were SDH tools that were not captured in this review because they were not mentioned in the reviewed articles. An additional search led us to identify other SDH screening tools that can be used in clinical settings such as the 211HelpSteps Program [96], EveryONE Project Social Needs Screening Tool [97], Institute of Medicine (IOM) Questionnaire [98], Your Current Life Situation Survey [99], and the Structural Vulnerability Assessment Tool [100]. As healthcare facilities modify and/or create their own SDH screening tools, it is likely that more screening tools will be available.

## 5. Conclusions

Socioeconomic factors are potent determinants of health. To attain a “health-creating” versus a “health-treating” system, healthcare needs to reassess where it is headed. It needs to shift gears and head from a downstream to an upstream direction. This calls for tackling the inequities where they are steepest and most concentrated—among at-risk, underserved, disadvantaged, and socially marginalized, whose day-to-day living and working conditions hinder them from achieving their best possible health. 

Creating health for all, especially among the vulnerable, necessitates an accessible, affordable, and a cohesive approach—one that combines social care into healthcare from a community perspective. Although not the sole strategy, integrating SDH in clinical settings is primarily carried out through patient-level SDH screenings. These allow for social needs to be identified and factored into the care of the individual—which would have otherwise persisted unrecognized and unattended. 

Gaps and challenges remain on how to incorporate SDH efficiently, sufficiently, and sustainably in established patient workflows as part of the standard of care. Apart from alleviating individual-level social needs and risks, there was little information from the current literature on whether healthcare facilities capitalize on the full potential of clinical SDH assessments nor on the extent to which SDH data were being utilized to invest in interventions directed at the root causes of health inequities at the community level.

There are several validated SDH screening tools available for use, actually used, and recommended for use in clinical settings. The question is not whether there is value added in assessing SDH in clinical settings. The challenge is in standardizing the collection of individual-level SDH data and incorporating them in EMRs and EHRs—in a way that the determinants being assessed align with the patient’s needs, and are the most relevant measures of health, and therefore, are also the strongest predictors of clinical outcomes. Available screening tools emphasize social needs and risks. However, they do not account for the protective factors and resources within the patient’s intra- and interpersonal social spheres that could counter the effects of adverse social factors. 

SDH assessments have yet to become the standard of care. However, several primary care and pediatric clinics, community health centers, safety net hospitals, and academic health centers are pioneering the feasibility of their inclusion in routine clinical practice. Nonetheless, data on their long-term impact toward the achievement of the Triple Aim are still lacking. Their full utility across the care continuum, beyond one-time assessments, and the scope of the concrete advantages they offer in clinical settings have yet to be fully explored and realized in advancing health equity. 

## Figures and Tables

**Table 1 ijerph-20-06873-t001:** SDH articles included in the literature review by study design and publication year.

Study Design	Reference Number	Number of Article *n* = 44	Percentage (%)	Publication Year *n* = 44
Quantitative Studies	[28,47,48,49,50,51,52,53,54,55,56,57,58,59,60,61,62,63,64]	19	42.2	2023 = 1 2022 = 11 2021 = 1 2020 = 4 2019 = 2
Qualitative Studies	[65,66,67,68,69,70,71,72,73]	9	22.2	2023 = 0 2022 = 2 2021 = 2 2020 = 3 2019 = 1 2018 = 1
Mixed Methods Studies	[74,75,76,77,78]	5	11.1	2022 = 2 2021 = 1 2020 = 2
Systematic Reviews	[26,79]	2	4.4	2022 = 1 2017 = 1
Theoretical Papers (Literature and/or Policy Reviews, Commentaries, Clinical Opinions)	[18,80,81,82,83,84,85,86,87]	9	20.0	2023 = 1 2021 = 1 2020 = 3 2017 = 2 2016 = 1 2008 = 1

**Table 2 ijerph-20-06873-t002:** Target patient population, conditions, and healthcare providers involved in integrating SDH in clinical settings based on a literature review.

Author, Year Published, Reference Number	Target Patient Population	Medical, Mental, and Behavioral Condition	Healthcare Provider Involved
	Summary: Integrating SDH in clinical settings focused primarily on vulnerable patients, especially those experiencing socio-economic challenges, material hardship, toxic stress, and inequities in health status and clinical outcomes	Summary: Integrating SDH in clinical settings focused largely on chronic diseases and/or mental health or behavioral issues vs. acute diseases.	Summary: Integrating SDH in clinical settings mainly involved healthcare providers in primary care settings.
Woolf et al., 2017 [18]	Low-income residents of inner-city neighborhoods with less education and social mobility; economically marginalized rural communities; racial/ethnic minority groups; Black infants and Black inmates with children	Diabetes, CVD, Cancer	Obstetricians
Parekh et al., 2022 [26]	Women experiencing food and/or housing insecurity, including those who were homeless	Impact of food and housing insecurity on CVD outcomes (MI, CHD, CHF) and stroke mortality	Physicians
Mahmood et al., 2023 [28]	National sample of women, aged 50–74	Food insecurity and biennial breast cancer screening behavior	Research team: University of Memphis School of public Health
Franz et al., 2022 [47]	Communities with significant economic and resource needs	None mentioned	Hospital-level SDH-specific activities in children’s hospitals
DeVetter et al., 2022 [48]	Physicians and patients who used AAFP’s Neighborhood Navigator (NN) tool with the Aunt Bertha/Find Help (AB/FH) community referral platform	Adverse SDH	Family physicians
Synovec and Aceituno 2020 [80]	Patients experiencing housing instability including homeless and refugees seen in an occupational therapy clinic	Diabetes, Stroke	Occupational therapy practice
Millender et al., 2022 [49]	Low-income, under-insured/uninsured; US migrants who spoke English as a second language and identified as African American, Black-Caribbean, Hispanic or Latinx, non-Hispanic White	Prostate cancer, Depression, PSA screening practices in diverse populations	Healthcare providers: Clinicians/physicians, nurses
Franz et al., 2019 [50]	20% of non-profit hospitals that responded to the 2015 American Hospital Association Annual Survey	Community-based efforts of non-profit hospitals to address opioid abuse	Research team: Ohio University
Begun et al., 2018 [65]	Hospitals	Impact of hospital-level community activities on population health and equity	Research team: University of Minnesota and University of South Florida, Tampa, Florida
Brennan et al., 2022 [66]	Pediatric practices that participated in AAP national QI project, “Addressing Social Health and Early Childhood Wellness”	Completion of SDH screenings and referrals from participating practices	10 pediatric practices Mid-central Indiana
Denny et al., 2019 [67]	Children, 0–4 years old, and their families	SDH and unintentional injuries	Core teams per participating practice: physician leader, nurse/nurse practitioner, medical assistant, and administrative staff/office manager
Ornelas et al., 2021 [74]	African Americans with CHF seen in a cardiology clinic	CHF	Providers and patients-cardiology clinic, ZSFG Hospital
Swamy et al., 2020 [51]	Primary caregivers, aged ≥18, of children aged 0–17 years seen for well-child/adolescent check-up; Pediatric clinical team	Unmet SDH needs and provider perspectives on SDH screening	Clinical team: pediatric residents, faculty, nurses, medical assistants, social workers, behavioral therapists, front office staff
Okafor et al., 2020 [75]	Underserved populations: migratory and seasonal agricultural workers, homeless, public housing residents, mostly females (58%), Hispanics (59%), and Blacks (30%)	Impact of food insecurity on diabetes, hypertension, heart disease, anxiety, depression, malnutrition, obesity, poor school performance, and hospitalizations	CEOs of healthcare organizations, chief medical officers, and pediatricians
Morris et al., 2020 [68]	Geriatric patients, aged ≥60, English-speaking, including proxies/caregiver of those with cognitive impairment	Screening for food insecurity and risks for malnutrition in an ED setting	ED staff; physicians, nurses, nursing assistants, and case managers
Kulkarni et al., 2023 [81]	Low-income families, mostly from Black and Hispanic communities, who were immunocompromised; vulnerable/disadvantaged; elderly or with disabilities; first-time mothers, women, and families; or with severe mental illness	CVD, MI, hypertension, diabetes, obesity, low birthweight, birth outcomes; HIV treatment in pregnancy; depression, functional issues; substance use (marijuana use), maternal smoking, and neighborhood violence	Clinicians/physicians
Berry et al., 2020 [69]	Patients of low-income status who were uninsured (30%), people of color and recent immigrants (90%), and Medicaid/Medicare recipients (49%)	Implementation and feasibility of SDH screening and referral programs in healthcare systems	Primary care providers, frontline clinical staff, Office of Population Health staff, and volunteer patient advocates
Brewster et al., 2020 [52]	Physician practices, national sample	Social risk screening, extent of participation in value-based payment models, and capacity for innovation	Physician practices
Cordova-Ramos et al., 2022 [76]	Level 2–4 NICUs across the US	Extent of standardized NICU SDH screenings in level 2 to 4 NICUs	Division chiefs, medical/clinical directors, and national sample
Horwitz et al., 2020 [77]	US hospitals	Scope and scale of investments in upstream social determinants made by health systems	626 healthcare systems, 917 hospitals (academic, profit, and non-profit)
Selvaraj et al., 2019 [53]	Low-income, racially diverse children aged 0–17 seen in well-child visits and their parents	Impact of unmet social needs on ACEs/toxic stress	Pediatric providers
Meyer et al., 2020 [54]	High-risk patients (≥2 ED visits in 12 months with ≥1 social needs using AHC screening tool); Latino neighborhoods (mostly foreign-born), limited English speakers (40%), living below poverty line (18%)); children aged 4–11 or ≥12 for asthma consult; Females (68%), Hispanic (82%); African American (14%), average household size of 3.6, average household income of USD 24,00	Asthma Substance use Alcoholism	Primary care physicians/ clinicians, practice administrators, volunteers who assisted patients on computer and health literacy issues, linked patients with community resources, collected data on performance improvement
Strenth et al., 2022 [55]	Patients (n = 581), aged 18–75 years, with Type 2 diabetes, seen at primary care clinics	SDH and interpersonal violence and effects on patients with type 2 diabetes	Family medicine physicians/primary care practitioners
Tung et al., 2022 [56]	Adult patients, of which 87% have never been screened for social isolation in clinical settings	Screening for social isolation in primary care settings and the impact of financial strain and intimate partner violence on social isolation	Clinicians
Gruß et al., 2021 [70]	CHCs across the US that adopted EHR-based SDH screening with no external implementation support	Factors facilitating the introduction and integration of EHR-based SDH screening in clinic workflows at CHCs	43 healthcare staff and professionals
Khatib et al., 2022 [57]	Patients in community hospitals	Testing of NowPow, a digital platform, for screening and referring patients for social needs across three community hospitals serving Chicago and its South Suburbs	Clinical teams: care managers, CHWs
Avallone et al., 2020 [71]	Seniors and elderly residing in apartments in a 23-story high-rise building	Nursing education training on the 4Ms framework: Matters–Medications– Mentation–Mobility to develop geriatric care and interprofessional competencies	Nursing faculty, 15 senior nursing students trained in 4M geriatric care framework, interprofessional team: pharmacy doctor, social workers, and CHWs
Montez et al., 2021 [58]	Children aged 0–18 seen at a primary care clinic and their parents	Food insecurity trends in an academic primary clinic	On-site care coordinator, 38 pediatric residents, and 15 physicians trained in food insecurity screening
Morone 2017 [82]	US pediatric populations	Available pediatric SDH screening tools	Clinicians/physicians, nurses, CHWs
Miller et al., 2022 [59]	Rural veterans, aged ≥50, with evidence of fracture risk seen at the VHA primary care, US Mountain West region	SDH and Osteoporosis screening behavior	Rural BHT: rheumatologist with osteoporosis expertise, physician assistant, pharmacist, nurses, support staff, and VHA primary care physicians
Fraze et al., 2021 [72]	Leaders and frontline staff from 29 healthcare organizations	Development and implementation of case-management-style programs to assist with social needs and referrals to community-based organizations	Hospital leaders, frontline staff
Power-Hays et al. 2020 [60]	Children with SCD whose respective families completed an SDH screening in a pediatric hematology clinic	Material hardships and percentage of ED visits among pediatric patients with SCD	Pediatric hematologists
Carter and Mazzoni 2021 [83]	Patients who were predominantly low-income, Black women, and Black babies living in St. Louis	Unacceptable pregnancy outcomes Depression, toxic stress, and unmet mental health needs	Physician-scientists, clinicians, obstetricians, and mental health professionals
Gerend and Pai 2008 [84]	American women who identified as African American and White	Social, economic, and cultural factors that contribute to Black–White disparities in breast cancer mortality	Clinicians/physicians, nurse, and patient navigators
Roland et al., 2017 [79]	Patients from medically underserved communities seeking care at FQHCs	Cancer-related CHW/patient navigator interventions in FQHCs, i.e., screening behavior for breast, cervical, and colorectal cancer and referral for screening	CHWs or patient navigators/lay health advisors/peer educators/promotoras
Hamilton et al., 2022 [61]	Pediatric patients aged 1–18 years admitted at the PICU for severe sepsis	Impact of census tract-level socioeconomic and neighborhood factors on PICU stay for severe sepsis	PICU staff
Tully et al., 2022 [73]	English and Spanish-speaking patients (n = 19) and healthcare team members (n = 11)	Patient and provider perspectives and recommendations on SDH screenings in maternity care	Physicians and staff Prenatal clinic
Fort et al., 2022 [62]	Pediatric patients with and without food insecurity	Screening for food insecurity and its documentation on medical charts	Pediatric clinic staff, medical assistants
Chukmaitov et al. 2022 [63]	Patients aged ≥18 years admitted to the Internal Medicine unit from communities around VCU with disproportionately high social needs and low life expectancy	SDH screening on food, housing, and transportation in a hospital setting	Internal Medicine unit staff, outreach community workers, VCU public health students
Hughes 2016 [85]	Low-income, under-resourced, at-risk populations; patients with depression	Diabetes Depression, ACEs Health Literacy	Family physicians, medical assistants, CHWs, and doulas
Quiñones and Hammad 2020 [86]	Patients with CKD	SDH and impact on onset and progression of CKD and ESRD	Primary care practitioners, physicians
Webb 2020 [87]	Patients with sickle cell disease, SCD	Impact of SDH on patients with SCD	Hematology practitioners, clinicians
Kim-Mozeleski et al., 2022 [64]	Socio-economically disadvantaged adult patients who were smokers, with many covered by Medicaid, Medicare, or uninsured	Impact of food insecurity, financial strain, transportation barriers, housing insecurity on smoking-related issues: COPD, CHF, CAD, Hypertension, and Diabetes	Care coordinators in a county hospital system
Massar et al., 2022 [78]	Clinic leadership, providers, and staff	Barriers and facilitators to implementing social needs screening and referral in pediatric primary care settings	Clinic leadership, providers, and staff from four pediatric ambulatory care clinics, New York City

Table abbreviations: AAFP—American Academy of Family Physicians; AAP—American Academy of Pediatrics; ACEs—Adverse Childhood Experiences; AHC-HRSN—Accountable Health Communities Health-Related Social Needs Screening; BHT—Bone Health Team; CAD—Coronary Artery Disease; CEOs—Chief Executive Officers; CHCs—Community Health Centers; CHD—Coronary Heart Disease; CHF—Congestive Heart Failure; CHWs—Community Health Workers; CKD—Chronic Kidney Disease; COPD—Chronic Obstructive Pulmonary Disease; CVD—Cardiovascular Disease; ED—Emergency Department; EHRs—Electronic Health Records; ESRD—End-Stage Renal Disease; FQHCs—Federally Qualified Health Centers; HIV—Human Immuno-deficiency Virus; MI—Myocardial Infarction; NLCHCs—Nurse-Led Community Health Centers; NICU—Neonatal Intensive Care Unit; PDSA—Plan–Do–Study–Act; PICU—Pediatric Intensive Care Unit; PSA—Prostate Specific Antigen; QI—Quality Improvement; SCD—Sickle Cell Disease; SDH—Social Determinants of Health; VCU—Virginia Commonwealth University; VHA—Veterans Health Administration; ZSFG—Zuckerberg San Francisco General.

**Table 3 ijerph-20-06873-t003:** SDH Screening Tools Used in Clinical Settings.

SDH Screening Tool * *n* = 29	Reference Number	Number of Article **
**AAP-NN—American Academy of Pediatrics’ Neighborhood Navigator** using Aunt Bertha/Find Help referral platform	[48,86]	2
**ACE** Survey—Adverse Childhood Experiences (10-item survey)	[55,85]	2
**ASK** Survey—**Addressing Social Key Questions for Health Questionnaire** (13-item screen for ACEs)	[53,87]	2
**AHC HRSN**—Accountable Health Communities Health-Related Social Needs Screening (26-item survey)	[56,69,76,81,86,87]	6
**Berkman-Syme Social Network Index, SNI** (on social isolation)	[56]	1
**BRFSS** Survey—Behavior Risk Factor Surveillance System (1–3-item food insecurity questions)	[26]	1
**CLEAR** Toolkit—Community Leadership on the Environment, Advocacy, and Resilience (four-step process for assessing patient vulnerability in a contextually appropriate and caring way)	[86]	1
**Family Needs Screening Tool** (28–33-item survey)	[87]	1
**HARK** Tool—Humiliation, Afraid, Rape, Kick (four-item survey)	[56]	1
**Health Begins Upstream Risks Screening Tool** (28-item survey)	[87]	1
**Health Leads Social Needs Screening Toolkit** (seven-item survey)	[63,81]	2
**HITS** Screening Tool—Hurt–Insult–Threaten–Scream (12-item survey)	[55]	1
**HVS**—Hunger Vital Sign (two-item survey)	[26,58,68,75]	4
**iHELP/iHELLP** Social History Tool—Income/Insurance–Hunger/Housing Conditions/Homeless–Education/Ensuring Safety–Legal Status, Literacy–Personal Safety (14–24-item survey)	[76]	1
**iScreen** Social Screening Questionnaire (46-item survey)	[87]	1
**MST**—Malnutrition Screening Tool (two-item survey)	[68]	1
**NASEM**—National Academies of Sciences, Engineering, and Medicine (one-item measure of financial strain)	[56]	1
**NHIS**—National Health Interview Survey, a CDC-NCHS ^1^ program	[26]	1
**NHANES**—National Health and Nutrition Examination Survey, a CDC NCHS ^1^ program (10-item food insecurity questions)	[26]	1
**NSHOS**—National Survey of Healthcare Organizations and Systems	[52]	1
**NowPow**—Digital screening and community referral platform for social needs, based on a modified PRAPARE	[57,69]	2
**PRAPARE**—Protocol for Responding to and Assessing Patient Assets, Risks, and Experiences (21-item survey)	[56,57,76,81]	4
**SEEK**—Safe Environment for Every Kid (16-item survey)	[51,67,76,87]	4
**WE CARE**—Well Childcare Visit, Evaluation, Community Resources, Advocacy, Referral, Education (10-item survey)	[51,76,87]	3
**WellRX**—Survey on four SDH domains: economic stability, education, neighborhood and physical environment, and food (11-item survey)	[85]	1
*Other Screening Tools Used:*		
**DDS**—Diabetes Distress Scale (17-item survey)	[55]	1
**PHQ-9**—Patient Health Questionnaire nine-item screening for depression (two-item survey)	[49]	1
**QILC**—Quality Improvement Learning Collaborative on performance measures	[67]	1

* This list included different SDH screening tools that were (1) actually used or evaluated in clinical settings; (2) cited in current SDH literature; or (3) recommended for use in clinical settings by researchers or experts. ** Although an article included in this literature review focused on or more SDH screening tools, or its modified version, the same article also cited other potential SDH screening tools for use in clinical settings. ^1^ CDC NCHS—Centers for Disease Control and Prevention National Center for Health Statistics.

**Table 4 ijerph-20-06873-t004:** Major reasons for integrating SDH in US clinical settings based on a literature review.

Primary Reason for Integrating SDH in Clinical Setting	Reference Number	Number of Article *
To implement an SDH screening process or program to identify unmet social needs, including food, housing, transportation, or material hardship, within the context of chronic diseases, mental health, and/or behavioral issues	[18,26,28,49,52,53,54,55,56,58,59,60,61,62,63,64,67,68,75,80,81,84,85,86,87]	25
To evaluate a current SDH screening process and/or seek perspectives on its implementation and challenges in patient workflows	[26,48,51,54,57,62,66,69,70,72,73,76,78,82,85]	15
To quantify/document hospital-level efforts in helping communities and/or linking patients with community resources and organizations	[47,50,52,65,72,77,79]	7
To identify and reduce health inequities in hospital processes	[74,83]	2
To improve hospital processes or patient workflows in integrating SDH assessments in clinical settings and increase cost-efficiency	[18]	1
To train future healthcare professionals in identifying and addressing SDH needs and risks	[71]	1

* Some articles included in this literature review may have multiple reasons for integrating SDH assessments in clinical settings.

## Data Availability

Not applicable.

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
