# Peer review of "Integrating Social Care into Healthcare: A Review on Applying the Social Determinants of Health in Clinical Settings"

_ijerph, 2023, doi:10.3390/ijerph20196873_

Round 1
Reviewer 1 Report
The study focuses on an important topic. Although there is a huge economical growth and incrersed wealth, today's world have many social and economical inequalities which causes inequality in accessing to the health services.
However, scientific impact is a little bit low beacuse it is a basic literature review. I would prefer to read a systematic review on the same topic.
The study title may be revised to tell the reader that it is a literature review.
Methods may be written longer to explaint the reader how the litereture was reviewed.
And a minor point, please just cite (1,2) in the end of the pragraph in Page 2 (first paragraph) instead of writing few times.
Good luck!
Author Response
We would like to thank all the reviewers and editors for their expertise and for dedicating their time in reviewing our paper and in providing valuable comments. We have carefully considered all the reviewers’ comments and suggestions and gave our best in sufficiently addressing all of them. It is our hope that the revised version of our manuscript meets MDPI's high standards. We welcome further constructive comments if any. Attached is our point-by-point response to all the comments and suggestions. The corresponding changes have been incorporated on our revised manuscript and were highlighted in yellow.
Please see the attachment.

Reviewer 2 Report
This is a timely, well-written article on an issue of growing and critical importance: the integration of social and health care systems, with social determinants of health as a connection point. I have only a few suggestions for improvement:
1. The phrase "the disadvantaged" on page 1, line 38 is off-putting.
2. In the results section, it would be useful to have an explicit transition sentence or two to alert the reader to the fact the data in the tables directly address the objectives of the study stated on page 5, starting on line 235.
3. I am not sure that the "unique article ID# column is useful to readers I am not clear what those numbers are or how readers might need/use them, since they do not correspond to numbered articles in the references.
4. The discussion section is excellent-- thoughtful, thorough, well-structured, insightful.
Author Response

(The authors gave the same response as above.)

Reviewer 3 Report
Overall, this manuscript provides very important information about the links between social determinants of health (SDOH) at the macro and micro levels. Need a clear purpose statement at the conclusion of the introduction. The study objectives were not found until lines 235-246, and they need to more clearly describe what they were aiming to identify and describe in the literature. There still remains confusion in the use of the concepts of health care and health. Also need a diagram of the steps in the approach used for the integrated literature review. The authors might consider adapting the PRISMA flow diagram.
Line 74: “1.1 The Lay of the Land: Social Determinants of Healthcare”. Do the authors mean social determinants of health rather than healthcare? The following paragraphs suggest they mean “health”.
Although there are some links between SD variables in section 1.1, the two major SDOH, education and food insecurity are the topical sentences.
Lines 122-124: Very short summary paragraph that may have more impact if used as a summary of the section on education.
Author Response

(The authors gave the same response as above.)
